# Survival outcomes and prognostic factors in children and adults with medulloblastoma from a Latin America country: A retrospective cohort

Michael Mallouh [1‡], Gabriel De la Cruz-Ku [2*‡], Renato Luque-Benavides [2], Bryan Valcarcel[3], Flavia Rioja[2], Martin Hemeryth[2], J. Smith Torres-Roman[4], Diego Chambergo-Michilot[2], Juan Haro Varas[5], Daniel Enriquez-Vera[6], Jhajaira M. Araujo [7], Rosdali Diaz-Coronado[5,8], Victor Castro Oliden [5]

**1** Department of Surgery, University of Massachusetts Chan Medical School, Worcester, Massachusetts, United States of America, **2** Universidad Cientifica del Sur, Lima, Perú, **3** George Washington University, Washington, DC, United States of America, **4** Escuela de Postgrado, Universidad Tecnológica del Perú, Lima, Peru, **5** Instituto Nacional de Enfermedades Neoplasicas, Lima, Peru, **6** Escuela Profesional de Medicina Humana, Universidad Privada San Juan Bautista, Lima, Peru, **7** Oncosalud-Auna, Lima, Peru, **8** Universidad Peruana Cayetano Heredia, Lima, Peru

‡ Co-First authorship
* gabrieldelacruzku@gmail.com

## Abstract

### Background

Few studies have evaluated the real-world outcomes of patients with medulloblastoma with contradictory results. Therefore, we aimed to compare the characteristics, survival outcomes, and prognostic factors between children and adults with medulloblastoma.

### Methods

We conducted a retrospective cohort study in a single academic center between 2000 and 2016. Patients were categorized into children and adolescents (≤19 years) and adults (>19 years). Overall survival (OS) and disease-free survival (DFS) were estimated with the Kaplan-Meier method. Prognostic factors were determined using Cox models.

### Results

In total, 173 patients were included (79 adults and 94 children). No differences were observed in clinical characteristics according to age groups. At 5 years, DFS was 36.88% in children and 50.99% in adults (p = 0.476). Prognostic factors of DFS in adults were radiotherapy (adjusted Hazard ratio [aHR]: 0.22; 95% confidence interval [CI]: 0.07–0.67) and performance status 2–4 (aHR: 3.60; 95% CI: 1.48–8.77); while in children was radiotherapy (aHR: 0.33; 95% CI: 0.11–0.96) and chemotherapy (aHR: 0.34; 95% CI: 0.14–0.84). The 5-year OS rate was 64.25% in children and 60.87%

**Data availability statement:** All relevant data are within the paper and its Supporting information files.

**Funding:** The author(s) received no specific funding for this work.

**Competing interests:** The authors have declared that no competing interests exist.

in adults (p = 0.447). In adults, prognostic factors of OS were histologic type (aHR: 6.11, 95% CI: 1.19–31.48, for anaplastic) and radiotherapy (aHR: 0.22, 95% CI: 0.07–0.71), while in children and adolescents, lower performance status (aHR: 2.43, 95% CI: 1.09–5.39, for performance status 2–4) was the only prognostic factor. Subgroup analyses revealed a trend toward improved DFS and significantly better OS among adults who completed chemotherapy, highlighting the importance of treatment adherence.

## Conclusions

Both populations demonstrated similar survival rates that were comparable to or lower than those reported in previous studies. In adults, performance status and radiotherapy were prognostic factors for DFS, while histologic type and radiotherapy for OS. In children, chemotherapy and radiotherapy were prognostic factors for DFS, and performance status was the only OS prognostic factor. These findings underscore the importance of optimizing treatment adherence and completing standard therapies, as well as improving staging practices, to enhance outcomes in real-world settings.

## Introduction

Medulloblastoma is an aggressive tumor arising in the posterior fossa and is classified as a grade IV tumor according to the World Health Organization. Medulloblastoma is the most common brain neoplasms in children, representing 15–20% of them, with an incidence rate (per million people) ranging from 4.5 to 6 [1,2]. In adults accounts for 1% of brain neoplasms with an incidence rate (per million people) ranging from 0.5 to 2 [3].

The prognostic classification divides the medulloblastoma in standard and high risk groups, the last one includes presence of metastases, postsurgical residual tumor more than 1.5 cm and large cell/anaplastic histology [4]. In addition, a consensus in 2010 indicated four subgroups from medulloblastoma: wingless (WNT), sonic hedgehog (SHH), group 3 and group 4, each of them with different biological and clinical characteristics, having group 3 the worst prognosis [5].

Since the introduction of chemotherapy to the standard treatment (surgery plus radiotherapy), patients diagnosed with medulloblastoma have increased their survival, mainly children [6–10]. In fact, some studies have shown a higher survival in children compared to adults [9,10]. This disparities in survival are associated to treatment. For instance, a study by Curran et al. showed better survival in children compared to adults, when both received radiotherapy alone, while in the comparison when both received radiotherapy plus chemotherapy showed no significant differences in outcomes [9].

In Peru, there are no epidemiological or clinical studies that evaluate the differences in the survival of children and adults with medulloblastoma. Therefore, we

sought to determine whether the characteristics and prognostic factors of survival differ according to age group in a cohort of patients diagnosed with medulloblastoma in Latin America.

## Materials and methods

### Study design and variables

We performed a retrospective cohort study at the National Institute of Neoplastic Diseases (INEN, in Spanish), Lima, Peru from 2000 to 2016. The INEN is a tertiary-care center and the national reference for neoplastic diseases with private/public mix operations [11]. In the public sector, it belongs to the health network of the Ministry of Health that covers over 80% of the total insured population in Peru [12].

Clinical information was obtained from patients' records, treated at the Department of Neurosurgery. Medical records were identified with the code C71 of the International Classification of Diseases – 10th edition. We included all patients with a new diagnosis of medulloblastoma, confirmed by pathology. We excluded patients with incomplete medical records, without pathologic confirmation of medulloblastoma, without completion of at least half of the prescribed chemotherapy cycles, and treated at another institution, and those who were lost to follow-up after the diagnosis. Patients were classified as children & adolescents (≤19 years) and adults (>19 years) as previous studies have done, which allows direct comparison with these cohorts [3,13].

Clinical variables were measured at cancer diagnosis. For adults, performance status was measured with the Eastern Cooperative Oncology Group (ECOG) score. Patients ECOG score 0–1 and score 2–4 were categorized as good and poor performance status, respectively [14]. For adolescents and children, Karnofsky [15] and Lansky [16] scale were used respectively, then the function level was categorized in numbers 0–5 to divide them in good and poor performance with scores 0–1 and 2–4, respectively (S1 Table). Other relevant measured variables were age, sex, histology type (desmoplastic/classic NOS and anaplastic), Metastases stage (M), presence of hydrocephalus, mesencephalic involvement, fourth ventricle involvement, midline involvement, tumor size after resection, risk classification of the Journal of Clinical Oncology JCO for medulloblastoma, radiotherapy, chemotherapy, and local relapse and distant relapse [17]. Radiotherapy was classified according to high a low dose for values of ≥56 Gy and <56 Gy, respectively. While for children and adolescents, craniospinal radiotherapy is administered at doses of 23 Gy for standard-risk patients and 36 Gy for high-risk patients [18].

### Outcomes

Our primary outcomes were overall survival (OS) and disease-free survival (DFS) at five years. We defined OS from medulloblastoma diagnosis until death from any cause or end of the study (June 30th, 2024). Data was accessed from June 1st, 2024 to June 30th, 2024. Similarly, DFS was defined from surgery date until progression of disease, local or distant relapse, death, or end of the study.

### Data analysis

We described clinicopathological and treatment features with frequencies and percentages and compared them with the Chi-square test. All quantitative variables were classified in categorical variables. OS and DFS was estimated with the Kaplan-Meier method and survival curves were compared with the Log-rank test. Median follow-up was estimated with the reverse Kaplan-Meier method. Age group was selected a priori as an effect modifier because previous research found different survival outcomes between both groups [4]. To address any potential bias, we included all the population available, no sample was calculated for our study. S1 File. Multivariate Cox regression models were fitted for the overall population and for each age group. The model included variables that were correlated with the age group and mortality, and variables that increased mortality in previous studies [3,17]. The final model included performance status,

type of histology, risk status, type of treatment, and age group (in the overall model). We used a 95% confidence interval (CI) for all the analysis and reported the Cox regression results with proportional Hazard Ratios (HR). Variables with a p-value < .05 were considered statistically significant.

### Ethical considerations

The present study was approved by the institutional review board of the INEN (INEN 24–30). Due to the retrospective The consent was waived by the ethics committee. The data was handled confidentially by the principal investigator, after having obtained the authorization to access the medical records by INEN review board. The study was funded by the authors, and no financial support was received from any financial institution or external entity. Patient information was handled with strict confidentiality and only for study purposes. Only the research team had access to the encoded data, which would not be disclosed.

## Results

### Baseline characteristics

Between 2000 and 2016, there were a total of 195 patients, from those 173 patients met the eligibility criteria to be included in the study, 79 (45.7%) adults and 94 (54.3%) children. The median age was 9 years for children and 26 years for adults. Most patients were men (60.1%), had a performance status 0–1 (77.1%), vermis involvement (93.0%), hydrocephaly (80.8%), underwent ventriculoperitoneal shunt (72.1%) and were free of metastases/M0 (94.4%) at diagnosis; and 55.2% were low risk. When stratified by age, there were no significant differences between baseline clinical characteristics between children and adult patients with medulloblastoma (Table 1).

### Treatment characteristics

All patients had surgery, except for one in the pediatric group. The type of surgery was reported in 164 patients and most of them had a complete surgery resection (59.8%). In total, 69.4% received chemotherapy, and 82.1% radiotherapy. Similarly, a total of 64.2% of the overall population received complete treatment (surgery plus chemotherapy plus radiotherapy). A total of 48.1% of adults and 46.8% of children completed the full course of chemotherapy (p = 0.865).. The median time to initiation of adjuvant therapy was 51 days, with radiotherapy and chemotherapy starting at a median of 53 and 150 days, respectively. Median time for children was 49 days, compared with 54 days for adults. Treatment characteristics had a similar distribution between children and adults. None of the patients in either cohort had documentation of neoadjuvant chemotherapy in their medical record. There were no significant differences between the pediatric and adult cohorts in distribution of patients receiving chemotherapy, radiation, or complete treatment. Among adults who had complete resection and non-complete resection, 80% and 75% received high dose radiotherapy, respectively. Pediatric and adult patients were placed on specific chemotherapy regimens at similar frequencies (Table 2).

### Disease-free survival and prognostic factors

The median follow-up was 47.73 months (Interquartile range (IQR): 18.00–89.53); 47.73 months (IQR: 18.00–84.90) for children and 40.37 months (IQR: 17.93–105.67) for adults. Both rate of both local and distant recurrence of medulloblastoma was also similar in the two groups, with both experiencing local recurrence (36.7% of adults vs 41.5% of pediatric patients) more commonly than distant recurrence (15.2% of adult patients vs 13.8% of pediatric patients). There was also a similar distribution of specific site of recurrence in pediatric and adult patients, with posterior fossa being the most common site of recurrence, followed by the spinal cord. 22.8% of adult patients had a recurrence at the posterior fossa vs 33.0% of pediatric patients; 8.9% of adult patients had a recurrence in the spinal cord versus 10.6% of pediatric patients.

 

**Table 1. Clinical features of medulloblastoma patients according to age group.**

| | Overall, n (%) | Adult, n (%) | Children, n (%) | P-value |
|---|---|---|---|---|
| No. of patients | 173 (100.0) | 79 (49.7) | 94 (54.3) | |
| Sex | | | | |
| Male | 104 (60.1) | 44 (55.7) | 60 (63.8) | 0.276 |
| Female | 69 (39.9) | 35 (44.3) | 34 (36.2) | |
| Performance status | | | | 0.154 |
| 0-1 | 128 (77.1) | 64 (82.1) | 64 (72.7) | |
| 2-4 | 38 (22.9) | 14 (17.9) | 24 (27.3) | |
| Missing | 7 | 1 | 6 | |
| Histologic type | | | | 0.054 |
| Classic/Desmoplastic | 69 (39.9) | 36 (45.6) | 33 (35.1) | |
| NOS | 88 (50.9) | 40 (50.6) | 48 (51.1) | |
| Anaplastic | 16 (9.2) | 3 (3.8) | 13 (13.8) | |
| Risk status | | | | 0.212 |
| Low risk | 91 (55.2) | 47 (60.3) | 44 (50.6) | |
| High risk | 74 (44.8) | 31 (39.7) | 43 (49.4) | |
| Missing | 8 | 1 | 7 | |
| Vermis involvement | | | | 0.890 |
| No | 11 (7.0) | 5 (6.7) | 6 (7.2) | |
| Yes | 147 (93.0) | 70 (93.3) | 77 (92.8) | |
| Missing | 15 | 4 | 11 | |
| Fourth ventricle involvement | | | | 0.475 |
| No | 79 (53.7) | 36 (50.7) | 43 (56.6) | |
| Yes | 68 (46.3) | 35 (49.3) | 33 (43.4) | |
| Missing | 26 | 8 | 18 | |
| Hydrocephaly | | | | 0.220 |
| No | 33 (19.2) | 12 (15.2) | 21 (22.6) | |
| Yes | 139 (80.8) | 67 (84.8) | 72 (77.4) | |
| Missing | 1 | 0 | 1 | |
| VPS | | | | 0.299 |
| No | 48 (27.9) | 19 (24.1) | 29 (31.2) | |
| Yes | 124 (72.1) | 60 (75.9) | 64 (68.8) | |
| Missing | 1 | 0 | 1 | |
| M stage | | | | NE |
| M0 | 151 (94.4) | 72 (97.3) | 79 (91.9) | |
| M1-4 | 9 (5.6) | 2 (2.7) | 7 (8.1) | |
| Missing | 13 | 5 | 8 | |

VPS: ventriculoperitoneal shunt.

NE: not evaluable.

In the study population, 5-year disease-free survival (DFS) was 43.2% (95% CI: 35.1–53.1), with 36.9% (95% CI: 26.6–51.2) in children and 51.0% (95% CI: 39.6–65.6) in adults (p = 0.476).

The multivariate analysis determined that performance status, histologic type, chemotherapy, and radiotherapy were prognostic factors of DFS at five years. Patients with performance status 2−4 (aHR: 1.95, 95% CI: 1.14–3.33, p = 0.015) and anaplastic histologic type (aHR: 2.53, 95% CI: 1.11–5.79, p = 0.027) had worse prognosis of DFS compared to those

**Table 2. Treatment and recurrence features of medulloblastoma patients according to age group.**

| | Overall | Adult | Children | P-value |
|---|---|---|---|---|
| No. of patients | 173 | 79 | 94 | |
| Surgery | 172 (99.4) | 79 (100.0) | 93 (98.9) | NE |
| Surgery resection | | | | 0.340 |
| Partial | 66 (40.2) | 28 (36.4) | 38 (43.7) | |
| Complete | 98 (59.8) | 49 (63.6) | 49 (56.3) | |
| Missing | 9 | 2 | 7 | |
| Chemotherapy | 120 (69.4) | 53 (67.1) | 67 (71.3) | 0.552 |
| Radiotherapy | 142 (82.1) | 67 (84.8) | 75 (79.8) | 0.391 |
| Surgery/Chemotherapy/ Radiotherapy | 111 (64.2) | 52 (65.8) | 59 (62.8) | 0.676 |
| Chemotherapy regimen | | | | |
| Vincristine/ Cisplatine/ Cyclophosphamide | 91 (75.8) | 43 (81.1) | 48 (71.6) | 0.195 |
| Vincristine/ Procarbazine/ Prednisone | 16 (13.3) | 3 (5.7) | 13 (19.4) | |
| Dacarbazine/Nitrosurea/Others | 13 (10.9) | 7 (13.2) | 6 (9.0) | |
| Site of recurrence | | | | |
| Local | 68 (39.3) | 29 (36.7) | 39 (41.5) | 0.536 |
| Distant | 25 (14.5) | 12 (15.2) | 13 (13.8) | 0.831 |
| Specific site of recurrence | | | | |
| Posterior fossa | 49 (28.3) | 18 (22.8) | 31 (33.0) | 0.176 |
| Pineal recess | 0 (0.0) | 1 (1.3) | 0 (0.0) | 1.000 [a] |
| Leptomeningeal | 2 (1.2) | 1 (1.3) | 1 (1.1) | 1.000 [a] |
| Spinal cord | 17 (9.8) | 7 (8.9) | 10 (10.6) | 0.696 |
| Spine | 4 (2.3) | 1 (1.3) | 3 (3.2) | 0.636[a] |
| Occipital lobe | 5 (2.9) | 3 (3.8) | 2 (2.1) | 0.514 |
| Distant bone metastasis | 1 (0.6) | 1 (1.3) | 0 (0.0) | 0.457 [a] |
| Cerebellum | 2 (1.2) | 1 (1.3) | 1 (1.1) | 1.000 [a] |
| Temporal lobe | 15 (8.7) | 7 (8.9) | 8 (8.5) | 0.935 |
| Bone marrow | 1 (0.6) | 1 (1.3) | 0 (0.0) | 0.457 [a] |
| Frontal lobe | 1 (0.6) | 1 (1.3) | 0 (0.0) | 0.457 [a] |
| Parietal lobe | 4 (2.3) | 1 (1.3) | 3 (3.2) | 0.626 [a] |
| Retroperitoneum | 1 (0.6) | 0 (0.0) | 1 (1.1) | 1.000 [a] |
| Distant lymph nodes | 1 (0.6) | 1 (1.3) | 0 (0.0) | 0.457 [a] |

NE: not evaluable, [a] Fisher exact test.

with performance status 0−1 and Classic/Desmoplastic histology, respectively. On the contrary, patients who received chemotherapy (aHR: 0.41, 95% CI: 0.23–0.73, p = 0.003) and radiotherapy (aHR: 0.24, 95% CI: 0.11–0.50, p < 0.001) had a better outcome (Tables 3 and 4).

In the analysis for each age group, radiotherapy remained as prognostic factor for both groups and was associated with a better 5 year-DFS (Figs 1–3). Performance status was found to be an independent prognostic factor only for adult patients (Table 3). Chemotherapy was found to be an independent prognostic only among pediatric patients (Table 3).

## Overall survival and prognostic factors

The median follow-up was 81.50 months (IQR: 47.40–126.97); 89.47 months (IQR: 60.63–135.67) for children and 72.13 months (IQR: 38.33–110.33) for adults. At 5 years, OS for the total population was 62.6% (95% CI: 55.3–70.8).

**Table 3. Prognostic factors of 5-years DFS and 5-years OS in patients with medulloblastoma.**

| Age group | Characteristics | 5-years DFS | | 5-years OS | |
|---|---|---|---|---|---|
| | | aHR (95% CI) | P-value | aHR (95% CI) | P-value |
| Total | Adults | 1.05 (0.63-1.75) | 0.849 | 1.70 (0.95-3.06) | 0.076 |
| | Men | 1.26 (0.73-2.19) | 0.405 | 1.74 (0.96-3.17) | 0.068 |
| | Performance status 2–4 | 1.95 (1.14-3.33) | 0.015 | 2.63 (1.44-4.80) | 0.002 |
| | High-risk status | 0.78 (0.48-1.28) | 0.323 | 0.89 (0.50-1.60) | 0.714 |
| | Histologic type* | | | | |
| | NOS | 1.11 (0.65-1.89) | 0.698 | 1.60 (0.86-2.98) | 0.142 |
| | Anaplastic | 2.53 (1.11-5.79) | 0.027 | 2.23 (0.90-5.57) | 0.085 |
| | Chemotherapy | 0.41 (0.23-0.73) | 0.003 | 0.52 (0.28-0.97) | 0.041 |
| | Radiotherapy | 0.24 (0.11-0.50) | <0.001 | 0.35 (0.17-0.75) | 0.007 |
| Adults | Men | 2.12 (0.91-4.96) | 0.083 | 1.59 (0.66-3.83) | 0.301 |
| | Performance status 2–4 | 3.60 (1.48-8.77) | 0.005 | 2.49 (0.92-6.78) | 0.073 |
| | High-risk status | 0.60 (0.26-1.41) | 0.243 | 1.05 (0.44-2.51) | 0.922 |
| | Standard-risk status | | | | |
| | Histologic type* | | | | |
| | NOS | 1.53 (0.70-3.34) | 0.290 | 2.40 (1.00-5.81) | 0.050 |
| | Anaplastic | 3.53 (0.73-16.99) | 0.116 | 6.11 (1.19-31.48) | 0.031 |
| | Chemotherapy | 0.47 (0.18-1.22) | 0.121 | 0.47 (0.18-1.20) | 0.113 |
| | Radiotherapy | 0.22 (0.07-0.67) | 0.008 | 0.22 (0.07-0.71) | 0.011 |
| Children | Men | 0.91 (0.44-1.88) | 0.799 | 1.89 (0.80-4.47) | 0.147 |
| | Performance status 2–4 | 1.40 (0.68-2.90) | 0.359 | 2.43 (1.09-5.39) | 0.029 |
| | High-risk status | 0.79 (0.40-1.54) | 0.482 | 0.67 (0.30-1.50) | 0.334 |
| | Standard-risk status | | | | |
| | Histologic type* | | | | |
| | NOS | 0.90 (0.40-2.00) | 0.793 | 0.95 (0.38-2.32) | 0.901 |
| | Anaplastic | 2.69 (1.00-7.28) | 0.050 | 1.51 (0.50-4.57) | 0.462 |
| | Chemotherapy | 0.34 (0.14-0.84) | 0.019 | 0.65 (0.26-1.66) | 0.370 |
| | Radiotherapy | 0.33 (0.11-0.96) | 0.041 | 0.54 (0.18-1.65) | 0.282 |

aHR: adjusted Hazard Ratio.

*: Classic/Desmoplastic as reference group.

OS was slightly higher in children compared to adults, although the difference was not statistically significant [64.3% (95% CI: 54.8–75.4) vs. 60.9% (95% CI: 50.3–73.6), p = 0.447]. When stratified by risk category, there were no significant differences in survival between children and adults. Among standard-risk patients, 5-year disease-free survival (DFS) was 30.5% versus 32.9% (p = 0.546), and overall survival (OS) was 57.1% versus 64.2% (p = 0.861) (Fig 2). Similarly, among high-risk patients, DFS was 24.1% versus 10.1% (p = 0.745) and OS was 68.1% versus 54.6% (p = 0.197) (Figs 4–7). Moreover, in a subgroup analysis of the adult population, patients who completed chemotherapy tended to have higher 5-year DFS rates (65.0% vs. 25.8%, p = 0.058) and significantly better OS rates (76.3% vs. 41.7%, p < 0.001) (Figs 8 and 9).

In the total study population, having performance status 2–4 was a prognostic factor for a worse 5-year OS, whereas receiving chemotherapy and radiotherapy were associated with a better prognosis (Tables 3 and 4). In adult patients, histologic type and radiotherapy were independent prognostic factors for overall survival. In children, the only independent prognostic factor for overall survival was performance status (aHR: 2.43, 95% CI: 1.09–5.39, p = 0.029, for ECOG 2–4) (Table 3).

**Table 4. OS and DFS at 5 years according to prognostic factors for medulloblastoma.**

| Age group | Characteristic | 5-years DFS | | 5-years OS | |
|---|---|---|---|---|---|
| | | % (95% CI) | P-value | % (95% CI) | P-value |
| Total | Performance status | | <0.001 | | 0.003 |
| | 0-1 | 55.56 (46.40-66.52) | | 69.95 (61.95- 78.96) | |
| | 2-4 | 13.15 (4.77- 36.26) | | 41.93 (28.10- 62.70) | |
| | Histologic type | | 0.030 | | 0.039 |
| | Classic/Desmoplastic | 50.63 (38.32- 66.88) | | 71.22 (60.2-84.2) | |
| | NOS | 41.57 (30.76- 56.18) | | 59.77 (49.94-71.54) | |
| | Anaplastic | 20.00 (6.10- 65.5) | | 42.86 (24.1-76.2) | |
| | Chemotherapy | | <0.001 | | < 0.001 |
| | No | 24.84 (12.52-49.27) | | 49.24 (37.18-65.21) | |
| | Yes | 48.3 (39.00-59.90) | | 68.41 (59.90-78.13) | |
| | Radiotherapy | | <0.001 | | < 0.001 |
| | No | 11.55 (3.27-40.84) | | 43.58 (28.89-65.74) | |
| | Yes | 48.82 (39.89-59.74) | | 66.44 (58.50-75.46) | |
| Adults | Performance status | | 0.001 | | 0.031 |
| | 0-1 | 63.85 (51.93-78.50) | | 67.8 (56.7- 81.1) | |
| | 2-4 | 0.00 | | 30.5 (12.2- 76.0) | |
| | Radiotherapy | | <0.001 | | < 0.001 |
| | No | 0.00 | | 26.67 (9.11-78.02) | |
| | Yes | 57.11 (45.0-72.5) | | 66.54 (55.37-79.96) | |
| Children | Performance status | | 0.124 | | 0.035 |
| | 0-1 | 47.67 (35.22- 64.53) | | 72.3 (61.4- 85.0) | |
| | 2-4 | 21.37 (8.25- 55.41) | | 48.2 (31.6- 73.7) | |
| | Chemotherapy | | <0.001 | | 0.061 |
| | No | 14.44 (2.51- 82.95) | | 54.44 (38.34-77.31) | |
| | Yes | 41.10 (29.57- 57.13) | | 67.98 (56.92-81.20) | |
| | Radiotherapy | | <0.001 | | 0.077 |
| | No | 17.17 (5.14- 57.39) | | 52.6 (34.4- 80.6) | |
| | Yes | 41.78 (30.1- 58.0) | | 66.66 (56.01- 79.33) | |

When stratified by age, patients with performance status 0–1 had a higher 5-year OS compared to performance status 2–4, in the total population study (Figs 10–12). Similarly, patients treated with chemotherapy or radiotherapy achieved better OS rates (Table 4).

Moreover, Patients with ECOG ≥2 had lower completion rates compared to ECOG 0–1 (35.7% vs. 51.6%, p = 0.283). Among those who completed chemotherapy, ECOG 0–1 patients showed significantly better 5-year OS (83.2% vs. 26.7%, p = 0.005) and a trend toward better DFS (27.6% vs. 0%, p = 0.721). In patients who did not complete chemotherapy, OS (45.5% vs. 31.7%, p = 0.841) and DFS (26.9% vs. 0%, p = 0.356) were higher in ECOG 0–1, though not statistically significant.

## Discussion

Differences in the outcomes and prognostic factors of pediatric and adult medulloblastoma patients remains controversial. Furthermore, characteristics and prognostic factor for medulloblastoma have not been evaluated in Latin-American patients. Our results showed that children and adults may have similar characteristics and outcomes in terms of OS and DFS; however prognostic factors changed according to age group.

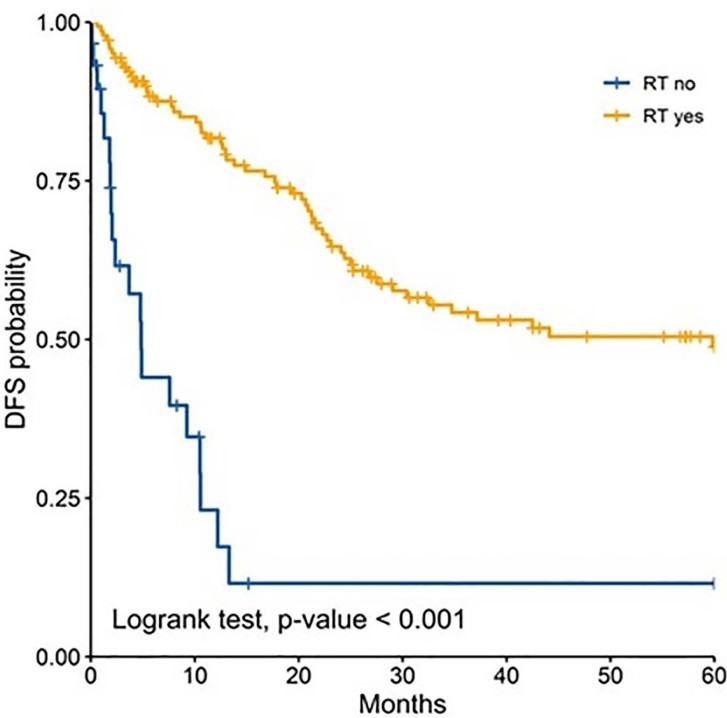

**Fig 1. Comparison of 5-years disease-free survival according to radiotherapy in all patients.**

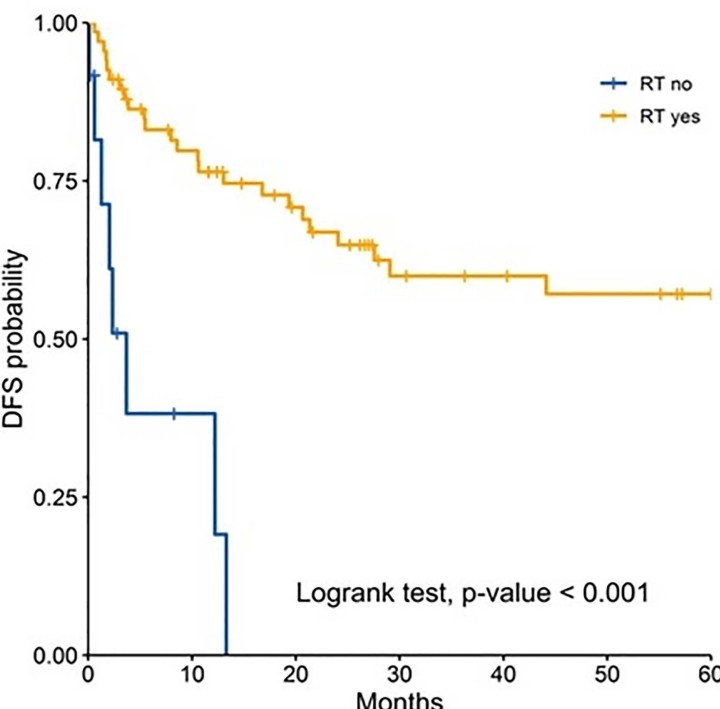

**Fig 2. Comparison of 5-years disease-free survival according to radiotherapy in adults.**

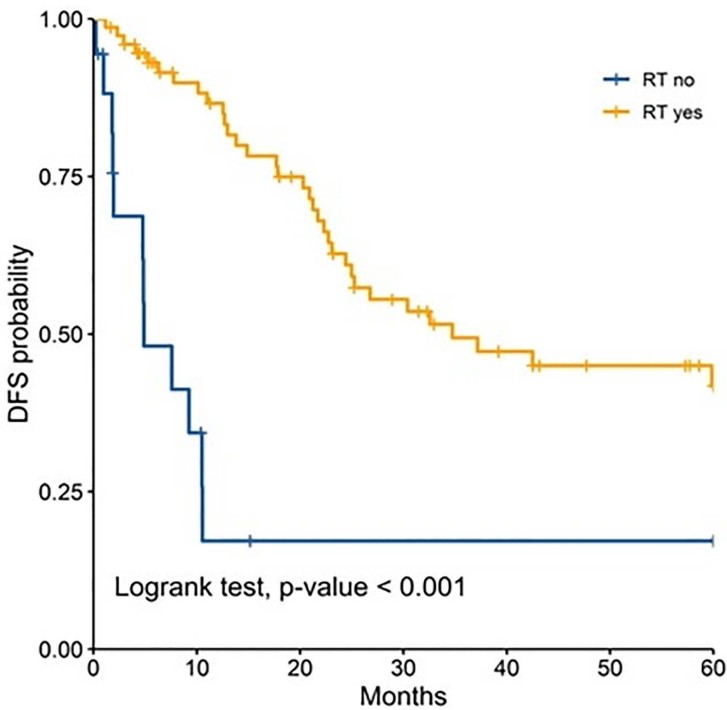

**Fig 3. Comparison of 5-years disease-free survival according to radiotherapy in children.**

We did not find differences in the clinical or treatment characteristics of children compared to adults. Likewise, Li et al. reported a similar distribution in both groups in regards of sex, race, histological type, tumor size, localization of tumor, surgery, and radiotherapy; however, they mentioned that distant metastasis were more frequent in children than in adults [17]. We did not find any difference in frequency of distant metastasis between children and adults. One possible explanation for the discordant finding in the two studies is that Li et al. had a greater number of overall patients (965 patients in Li et al. study versus 173 in ours), as well as a higher proportion of patients with distant metastasis at diagnosis (11.7% in Li et al. study versus 5.6% in our study). Other studies report percentages of children with metastatic medulloblastoma at time of diagnosis as high as 35% [19,20]. Further research is needed to better understand the factors that are contributing to this population having a higher percentage of local disease, specifically focusing on both histologic characteristics that preclude slower growth and rate of metastasis.

Our results showed that children and adults have similar characteristics, survival outcomes, and unique prognostic factors in for OS and DFS. In the overall study population, we found worse OS and DFS prognosis with a performance status score of 2–4, and better OS and DFS prognosis with receipt of either chemotherapy or radiotherapy. We did find that prognostic factors differ in adult versus pediatric patients. When examining the pediatric cohort alone, we found that the only prognostic factor was worse overall survival with a performance status of 2–4. Like the overall study cohort, we found that specifically pediatric patients had a better DFS prognosis with receipt of chemotherapy and/or radiotherapy. In adults, we found that a performance status of 2–4 was associated with worse overall survival and DFS prognosis, while radiotherapy and histologic type were associated with improved OS prognosis. When stratified by functional status, patients with ECOG 0–1 had higher chemotherapy completion rates and demonstrated significantly better OS, as well as a trend toward improved DFS, compared with ECOG ≥2. These findings suggest that performance status influences both treatment adherence and survival outcomes, highlighting the importance of optimizing supportive care to enable chemotherapy

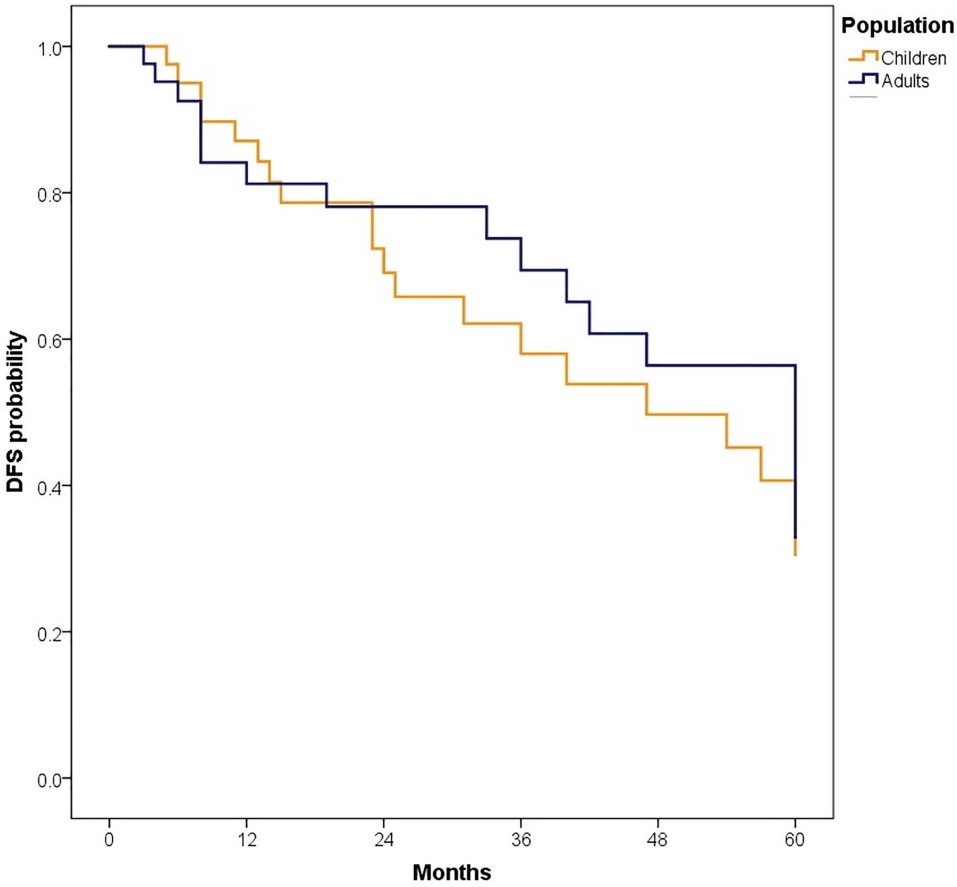

**Fig 4. Comparison of 5-year disease-free survival in standard-risk patients.**

completion. In the adult population we also found that NOS and anaplastic histologic types were associated with worse OS prognosis. Adults and children have unique prognostic factors for both OS and DFS, future studies are needed to examine histopathologic and molecular characteristics in both populations to better explain these findings.

On the other hand, in this study the 5-year DFS was 36.9% in children and 51% in adults. The rate in children rises up to 41.8% for children who received radiotherapy and 41.1% for children who received chemotherapy; however, is lower compared to a study that evaluated average-risk pediatric patients treated with craniospinal irradiation and chemotherapy, where 5-year DFS was 75.0% [21]. In adults, Mallick et al. has reported a similar 5-year DFS (50.7%), although their study only included 31 patients treated with surgery followed by craniospinal irradiation and adjuvant chemotherapy [22]. In addition, a systematic review that included patients between 15–39 years, reported that DFS ranged from 49% to 89% [23].

Furthermore, prognostic factors of DFS included radiotherapy and chemotherapy. In children, 5-years DFS seems to be not affected by time between surgery and RT initiation, RT duration, concomitant chemotherapy during radiotherapy, or maintenance chemotherapy compliance [24,25]. These differ from those in a study who found significant improvement in event free survival in patients who completed RT within 50 days than those who completed it in greater than 50 days [24]. However; DFS was better for patients who received all cycles of maintenance chemotherapy [25]. Other studies have found that findings are not in agreement with other studies examining retrospective outcomes data on pediatric patients with medulloblastoma, that found no difference in DFS with varying compliance [24,25].

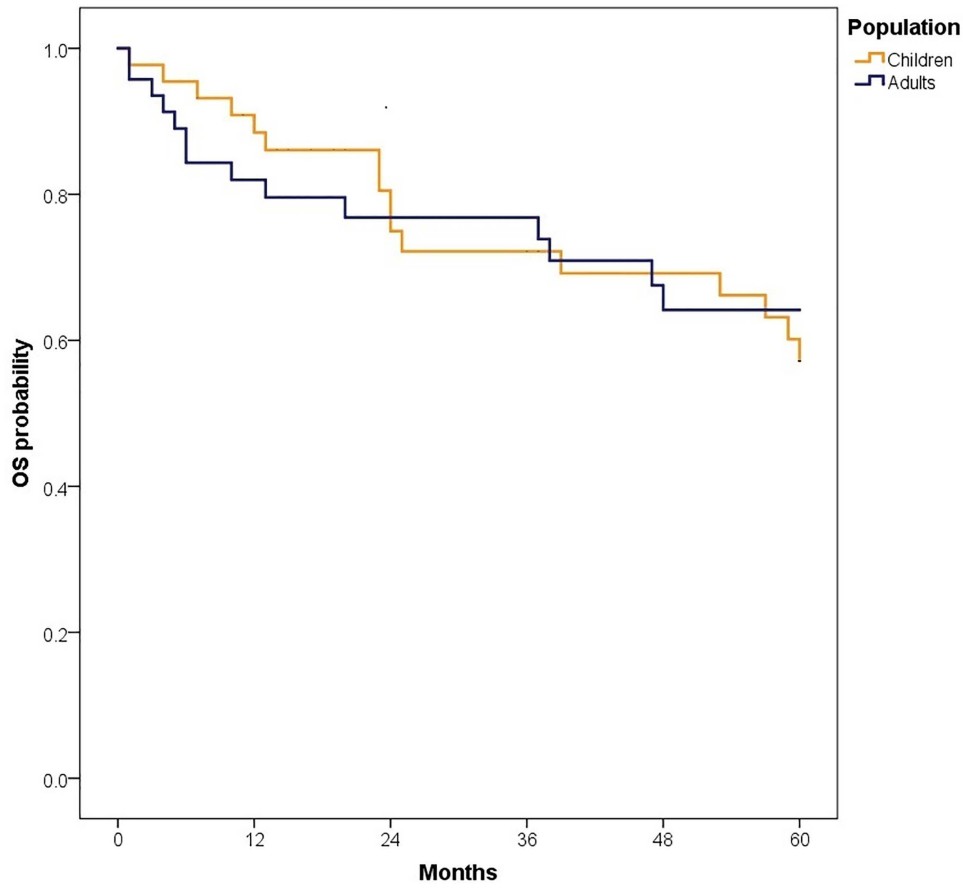

**Fig 5. Comparison of 5-year overall survival (OS) in standard-risk patients.**

Radiotherapy remained as an independent prognostic factor for improved DFS in across all age groups. It has been demonstrated to be a cornerstone in the treatment of medulloblastoma. For instance, Liu et al. reported in a systematic review that included 8500 patients that radiotherapy was associated with 69% improvement of event-free survival [26]. Similarly Seidel et al. reported survival benefits of this adjuvant therapy with further stratification based on risk classification [27]. It is important to acknowledge that radiotherapy is recommended to be administered in a timely fashion, as delayed initiation of craniospinal irradiation has been associated with worse survival outcomes [28].

In our cohort, local recurrence predominated in both children and adults, differing from international patterns [29]. In children, distant recurrence was similar to other low- and middle-income countries (13.8% vs 12.6%), but local recurrence was higher (41.5% vs 24.3%) [30–32]. These differences may reflect racial or molecular heterogeneity, limited molecular classification, and incomplete follow-up. Our findings highlight the need for prospective, multi-institutional studies with systematic imaging and molecular profiling to better characterize recurrence patterns.

We found that the 5-year OS rate in children was 64.3%. This result is higher in comparison to a study by Massimino et al., which showed an OS of 41% at 5 years post-diagnosis, and to what Karoly et al. reported in 80 children (5-year OS 53.8%) [33]. These slight differences may be explained by different definitions of "children," for instance, OS in the study by Massimino et al. and Karoly et al. were calculated for patients younger than 10 years and 15 years, respectively, while in this study the pediatric 5-year OS calculation included patients up to 19 years in age. In adults, 5-year OS in this study was 60.87%. This is similar to results showed by Lai R. et al. where 5-year observed survival was 64.0%, considering

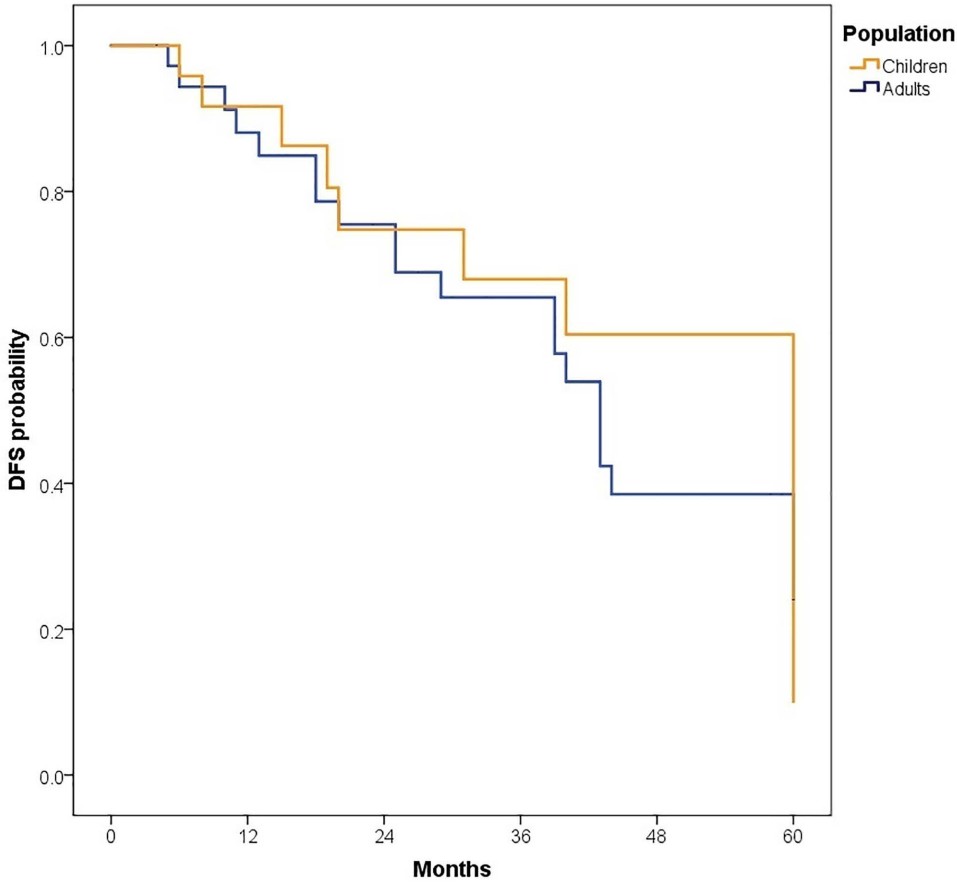

**Fig 6. Comparison of 5-year disease-free survival in high-risk patients.**

adults patients as those ≥18 years [10]. However, the adult 5-year OS in this study is slightly lower to rates achieved in UK where the 3- and 5-year OS rate were 79.5% and 70.7%, respectively [34]. In a study that evaluated patients from the SEER database and considered children patients aged between 4 and 19 years and, adults patients ≥20 years, the 5-year OS was 75.5% for children, and 74.2% for adults [17].

Although there is scarce literature in disparities on medulloblastoma survival, the decreased survival rate in our population may be attributed to multiple factors, but primarily to the high-risk disease with the majority of patients undergoing only subtotal resection, leaving large residual tumors. This is explained by patients seeking medical care at advanced stages of the disease, similar to other malignancies in our setting [35–37]. Contributing factors include limited access to treatment, delayed diagnosis, long waiting times for clinic appointments, extended referral times from other institutions due to an overwhelmed healthcare system, and social and economic barriers. [38]. Extent of CNS disease at diagnosis, and amount of residual disease after surgery have been found in other studies to be prognostic factors in medulloblastoma [39]. It is reasonable to expect that the lack of reliable interaction with healthcare services that our study population experiences would precipitate failure to detect early disease and thus worse outcomes with more extensive CNS involvement and partially resectable disease.

Another important factor potentially associated with worse overall survival is delayed initiation of adjuvant therapy. Radiotherapy is generally recommended to begin within four weeks after surgery, and chemotherapy within eight weeks. In our cohort, substantial delays beyond these recommended timeframes were observed for both modalities, primarily due

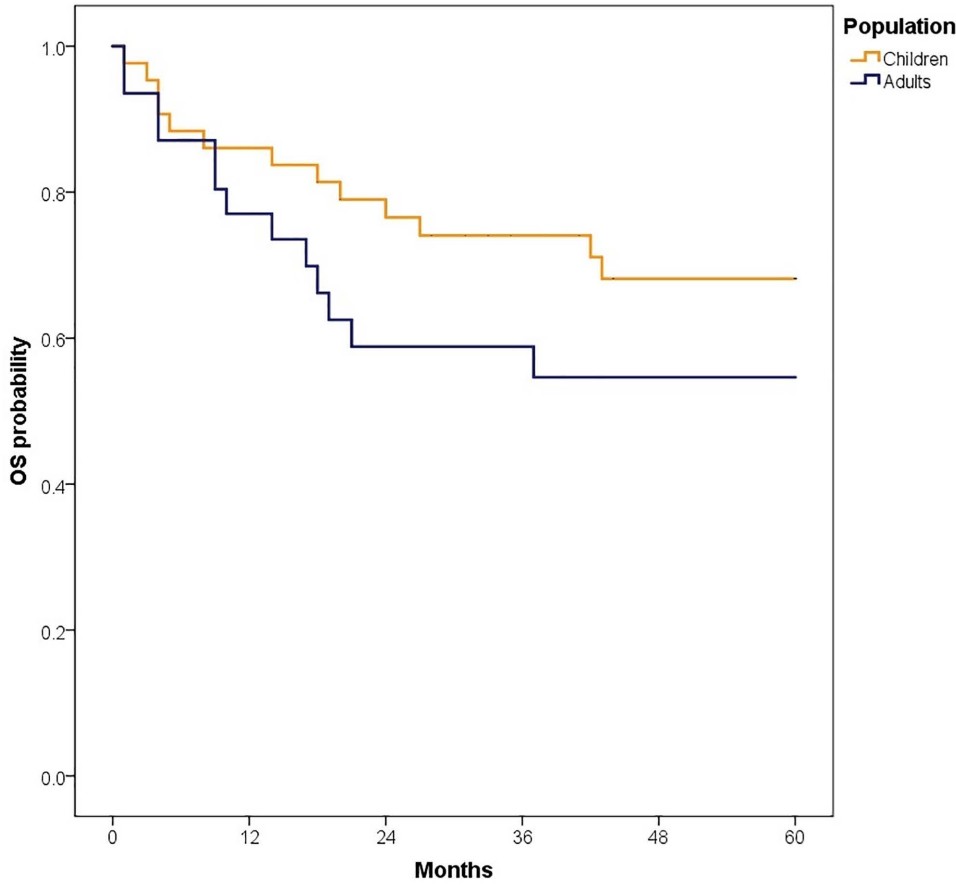

**Fig 7. Comparison of 5-year overall survival in high-risk patients.**

to high patient demand and limited institutional capacity, despite the availability of advanced equipment and radiotherapy techniques [40,41]. These findings highlight the challenges currently faced in our setting and underscore the urgent need to shorten the interval between surgery and adjuvant therapy.

The 5-years OS in children in our study who received radiotherapy was 66.7%, which is slightly lower compared to a study by Taylor et al., where the impact of radiotherapy parameters was analyzed in 179 children and a 5-year OS of 70.7% was reported [24]. Their study also showed that patients who completed RT in less than 50 days had a statistically significant improvement in both overall survival and event free survival than patients who took greater than 50 days to complete RT [24]. These findings should be further evaluated due to possible distinct effects among different populations. For instance, Li Q et al. reported that radiotherapy was associated with a reduced hazard of mortality for children (HR 0.36, 95% CI 0.22–0.60) and adults (HR 0.47, 95% CI 0.26–0.84) [17].

Performance status was a prognostic factor of 5-year DFS in adults and it was the only prognostic factor for 5-year OS in children. Although, in both groups, adults and children, 5-years OS was better in patients with a good functional status (performance status 0–1 v ECOG 2–4). A study that included patients between 16–62 years, also reported that patients with performance status 3 had increased hazard (HR 2.7, CI 1.63–4.55) compared to patients performance status 0–2 [42]. In addition, a post operative performance status more than 2 has been associated with a detriment in event-free survival and OS, in patients older than 18 years [43,44].

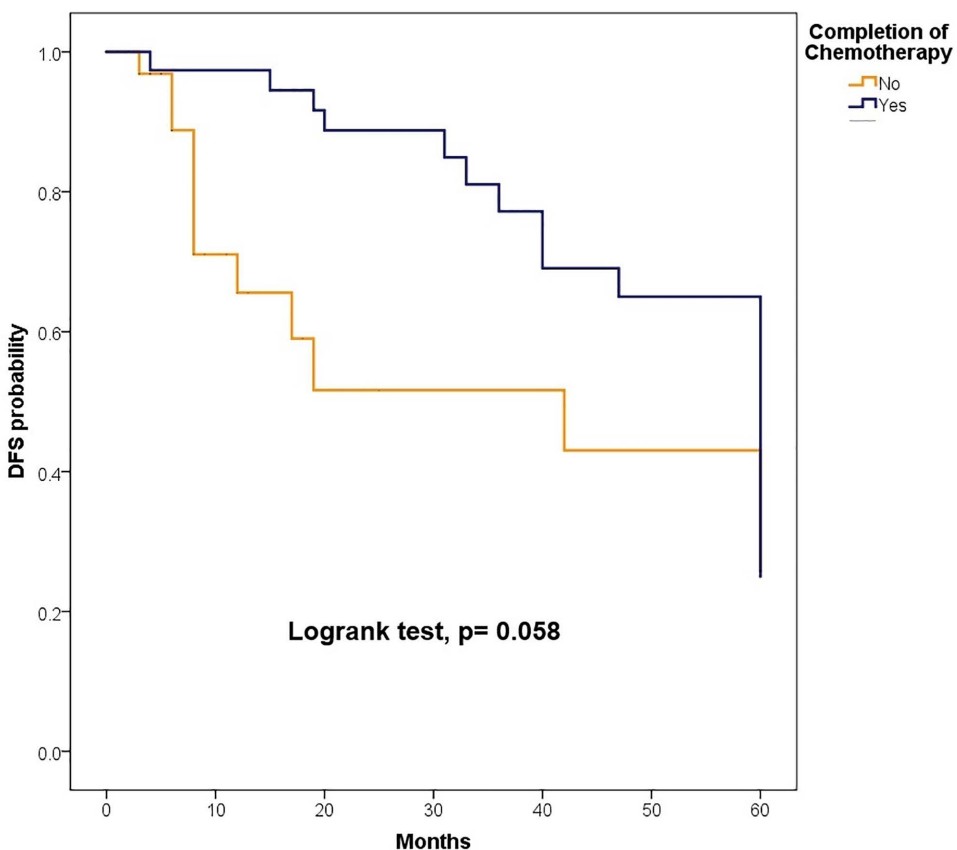

**Fig 8. Comparison of 5-year disease-free survival according to the completion of chemotherapy.**

Recent European guidelines (EURACAN, 2019) recommend chemotherapy for all adult medulloblastoma patients regardless of risk group, reflecting general scientific consensus [45]. In our cohort, 67.1% of adults received chemotherapy, but regimens were heterogeneous and adherence was suboptimal, with only 48.1% completing the prescribed cycles. Although lower performance status could potentially affect treatment tolerance, adults generally had better performance status than children (ECOG 0–1: 81.1% vs. 72.7%), and the difference was not statistically significant. The adult population tended to receive slightly less chemotherapy than children (67.1% vs. 71.3%, p = 0.552) but had similar rates of completing the full course of cycles (48.1% vs. 46.8%, p = 0.865). Subgroup analysis revealed a trend toward improved DFS and significantly better OS among adults who completed chemotherapy, emphasizing the importance of treatment adherence, consistent with previous studies, and highlighting the need to guide our practice according to international guidelines [45,46]. The absence of DFS improvement in the overall cohort is likely attributable to incomplete adherence rather than baseline functional status or age-related comorbidities.

Contributing factors included delayed or limited access to specialized care, financial constraints, suboptimal condition at presentation, poor adherence, and systemic delays [47,48]. Presenting recent trial data has helped improve adherence and completion. These findings highlight the need to enhance access, adherence, and completion of chemotherapy, and future prospective studies are warranted to clarify the potential survival benefit in adults when treatment is fully administered.

This study has some limitations. We could not analyze whether some aspects of treatment, such as the impact of each chemotherapy regimen as prognostic factor. The study was conducted with data from the main Peruvian Cancer Institute;

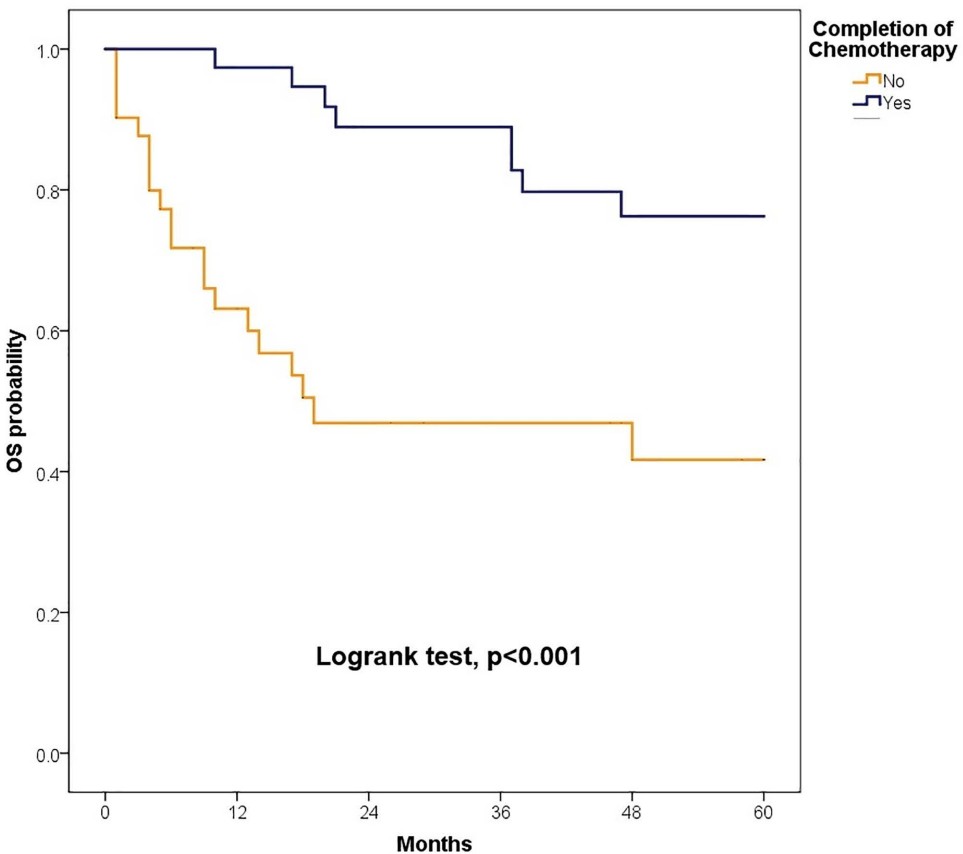

**Fig 9. Comparison of 5-year overall survival according to the completion of chemotherapy.**

therefore, these should be taken with caution when generalization or extrapolation of our findings to other populations. Assessing a longer follow-up may reveal other prognosis factors. The age cutoff used to define pediatric versus adult patients was based on precedent publications. Although somewhat arbitrary, it does not affect the internal validity of our analyses and provided a consistent framework for evaluating our population. The pediatric population may be under-represented in our cohort, as there are at least two to three national institutions for children with this disease, compared to only one for adults. Prospective multi-institutional studies or population-based registries are needed to confirm our findings. We did not classify medulloblastoma into molecular subgroups due to the unavailability of molecular testing in our population, which is limited by resources. Molecular classification could provide further insights into survival outcomes, as each subgroup has a distinct prognostic profile. This limitation is common in low-income countries, highlighting the need for new policies and increased funding to support complete molecular classification, improve personalized therapies, and ultimately achieve better survival outcomes. There is potential bias in the final staging due to limited access to MRI or CT scans and the restricted performance of lumbar puncture for several reasons. MRI availability is limited in many institutions in our country, including some national high-complexity hospitals. Lumbar puncture was also not consistently performed, which likely contributed to understaging and may have affected survival estimates. Some patients underwent surgery at non-oncologic centers and continued treatment at our institution, where lumbar puncture or MRI had not been performed. Another limitation of our study is the incomplete histologic subtyping of medulloblastoma cases, as many were classified as NOS. This restricts the ability to analyze histology-specific outcomes and may obscure differences between

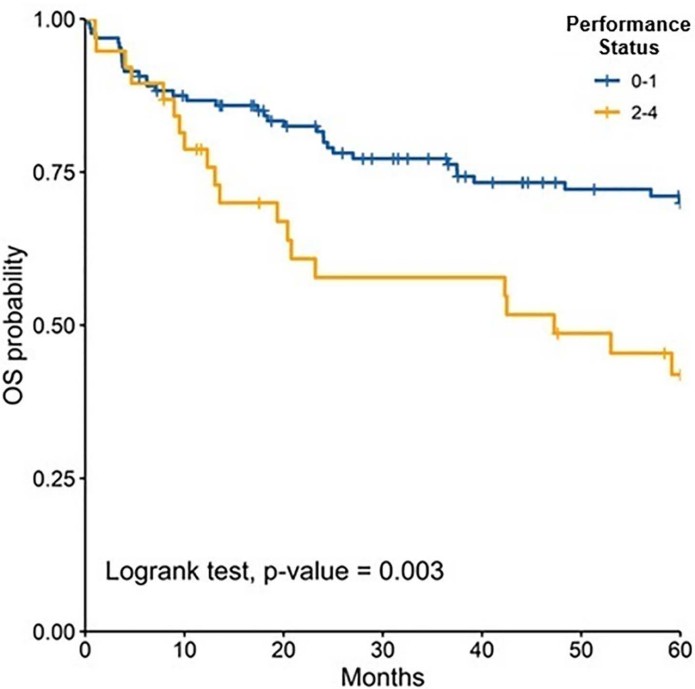

**Fig 10. Comparison of 5-years overall survival according to ECOG in all patients.**

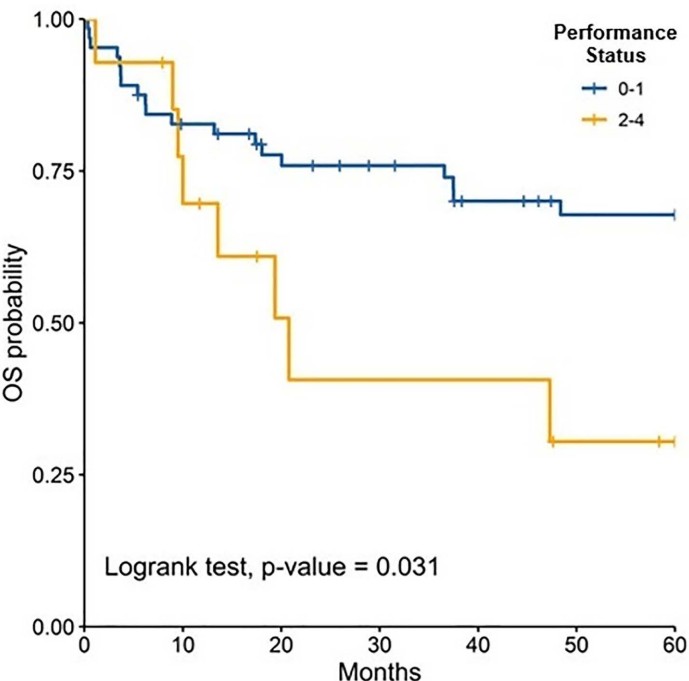

**Fig 11. Comparison of 5-years overall survival according to ECOG in adults.**

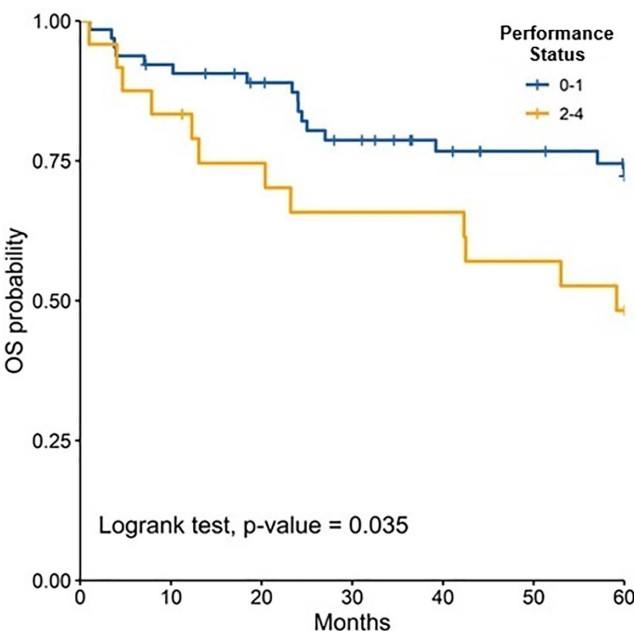

**Fig 12. Comparison of 5-years overall survival according to ECOG in children.**

subtypes. Future studies with prospective collection of detailed histologic and molecular data, in accordance with the 2021 WHO CNS classification, are needed to better characterize outcomes by subtype [49]. Additionally, the absence of a standardized medulloblastoma protocol in our country and the retrospective design of our study contribute to this limitation. Nevertheless, these findings provide valuable evidence to inform the future implementation of policies and optimize current practices in our country. Further research in disparities in this neoplasm is needed due to possible different molecular biology and behavior among different races and regions.

## Conclusion

Both Latin American populations according to age group demonstrated similar survival rates that were comparable to or lower than those reported in previous studies.; however, prognostic factors were unique for each population. In adults, performance status, chemotherapy, and radiotherapy were prognostic factor for 5-years DFS whereas histologic subtype and radiotherapy were prognostic factors for 5-years OS. In children, only chemotherapy and radiotherapy were statistically significant for 5-years DFS, and performance status was the only independent prognostic factor of 5-years OS. Further research is needed in terms of molecular biology and disparities to understand possible different behavior and outcomes of medulloblastoma.

## Supporting information

**S1 File. De-identified Excel dataset containing clinical and pathologic information on Latin American patients diagnosed with medulloblastoma.**
(XLSX)

**S1 Table. Comparison of the Lansky, Karnofsky, and ECOG performance status according to the function level of pediatric, adolescents, and adult patients.**
(DOCX)

## Acknowledgments

The authors thank the Universidad Científica del Sur for their support in the publication of this research/project.

## Author contributions

**Conceptualization:** Michael Mallouh, Gabriel de la Cruz-Ku, Renato Luque-Benavides, Bryan Valcarcel, Flavia Rioja, Martin Hemeryth, J. Smith Torres-Roman, Diego Chambergo-Michilot, Juan Haro Varas, Daniel Enriquez-Vera, Jhajaira M. Araujo, Rosdali Diaz-Coronado, Victor Castro Oliden.

**Data curation:** Gabriel de la Cruz-Ku, Flavia Rioja, J. Smith Torres-Roman, Diego Chambergo-Michilot, Juan Haro Varas, Victor Castro Oliden.

**Formal analysis:** Gabriel de la Cruz-Ku, Renato Luque-Benavides, Rosdali Diaz-Coronado.

**Investigation:** Michael Mallouh, Gabriel de la Cruz-Ku, Renato Luque-Benavides, Flavia Rioja, Martin Hemeryth, Diego Chambergo-Michilot, Juan Haro Varas, Daniel Enriquez-Vera, Jhajaira M. Araujo, Rosdali Diaz-Coronado.

**Methodology:** Michael Mallouh, Gabriel de la Cruz-Ku, Renato Luque-Benavides, Bryan Valcarcel, Flavia Rioja, Diego Chambergo-Michilot, Jhajaira M. Araujo, Rosdali Diaz-Coronado, Victor Castro Oliden.

**Project administration:** Michael Mallouh, Gabriel de la Cruz-Ku, Renato Luque-Benavides, Jhajaira M. Araujo, Rosdali Diaz-Coronado.

**Resources:** Gabriel de la Cruz-Ku.

**Software:** Gabriel de la Cruz-Ku, Flavia Rioja.

**Supervision:** Michael Mallouh, Gabriel de la Cruz-Ku, Bryan Valcarcel, Flavia Rioja, J. Smith Torres-Roman, Daniel Enriquez-Vera, Rosdali Diaz-Coronado, Victor Castro Oliden.

**Validation:** Michael Mallouh, Gabriel de la Cruz-Ku, Renato Luque-Benavides, Flavia Rioja, Martin Hemeryth, Diego Chambergo-Michilot, Juan Haro Varas, Daniel Enriquez-Vera, Jhajaira M. Araujo, Rosdali Diaz-Coronado, Victor Castro Oliden.

**Visualization:** Michael Mallouh, Gabriel de la Cruz-Ku, Renato Luque-Benavides, Bryan Valcarcel, Flavia Rioja, Martin Hemeryth, J. Smith Torres-Roman, Diego Chambergo-Michilot, Juan Haro Varas, Daniel Enriquez-Vera, Jhajaira M. Araujo, Rosdali Diaz-Coronado, Victor Castro Oliden.

**Writing – original draft:** Michael Mallouh, Gabriel de la Cruz-Ku, Renato Luque-Benavides, Bryan Valcarcel, Flavia Rioja, Martin Hemeryth, J. Smith Torres-Roman, Diego Chambergo-Michilot, Juan Haro Varas, Daniel Enriquez-Vera, Jhajaira M. Araujo, Rosdali Diaz-Coronado, Victor Castro Oliden.

**Writing – review & editing:** Michael Mallouh, Gabriel de la Cruz-Ku, Renato Luque-Benavides, Bryan Valcarcel, Flavia Rioja, Martin Hemeryth, J. Smith Torres-Roman, Diego Chambergo-Michilot, Juan Haro Varas, Daniel Enriquez-Vera, Jhajaira M. Araujo, Rosdali Diaz-Coronado, Victor Castro Oliden.

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
