## [Decision Letter · Decision Letter 0]

25 Jul 2025

Dear Dr. de la Cruz-Ku,

Thank you for submitting your manuscript to PLOS ONE. After careful consideration, we feel that it has merit but does not fully meet PLOS ONE’s publication criteria as it currently stands. Therefore, we invite you to submit a revised version of the manuscript that addresses the points raised during the review process.

We look forward to receiving your revised manuscript.

Kind regards,

Michael C Burger, M.D.

Academic Editor

PLOS ONE

Journal Requirements:

Reviewers' comments:

Reviewer's Responses to Questions

**Comments to the Author**

1. Is the manuscript technically sound, and do the data support the conclusions?

Reviewer #1: Yes

Reviewer #2: Partly

2. Has the statistical analysis been performed appropriately and rigorously?

Reviewer #1: Yes

Reviewer #2: Yes

3. Have the authors made all data underlying the findings in their manuscript fully available?

Reviewer #1: Yes

Reviewer #2: Yes

4. Is the manuscript presented in an intelligible fashion and written in standard English?

Reviewer #1: Yes

Reviewer #2: Yes

Reviewer #1: The present paper offers clinical and follow-up data which appear correctly presented with survival analysis (OS/DFS curves, Cox models) executed correctly for a retrospective study.

While risk status by age is reported in the tables no separate Kaplan-Meier curves stratified by risk are presented and should be added for visual clarity; furthermore a dedicated survival analysis for pediatric versus adults standard- and high-risk groups could give more insight into outcomes

The identification of radiotherapy and performance status as significant prognostic factors are supported by the data presented. However the lack of molecular subgrouping, which has now become a central aspect in medulloblastoma management, limits the possibility to draw definite conclusions but it is correctly and explicitly acknowledged by the authors as a key limitation of the study.

Recent European guidelines (from EURACAN) recommend chemotherapy for all adult medulloblastoma patients regardless of risk group aligning with the general scientific consensus, this should be cited and discussed, more than half the adults treated in the institution received chemotherapy but without the authors commentary it’s not clear if treatment practice is aligning with the evolving therapeutic standards.

67.1% of adult patients received chemotherapy despite heterogenous regimens, while recent evidence suggest that adherence to multi cycle chemotherapy, regardless of the protocol timing or composition, provides benefit on survival, the authors should address possible reason for not finding and improved DFS in the adult population receiving chemotherapy (complications related to chemotherapy? Co-morbidity related to age? Sample size?)

Could lower performance status, suggesting a baseline worse functional status, be reflecting of an overall worse tolerance to treatment and, if so, can it be linked to the lower percentage of adults undergoing chemotherapy in comparison to the pediatric group? It might be helpful to discuss if the adult patients who completed their planned chemotherapy cycles showed an improvement in survival compared to those that did not (if data for those patients is available). It’s interesting to note that despite adults having better performance status at diagnosis (ECOG 0-1 82.1% adults vs. 72.7 children) they received chemotherapy less often (67.1% vs. 71.3%) indicating that performance status alone doesn’t account for treatment differences, an analysis of chemotherapy completion in regard to ECOG status and its impact on survival could tell if functional status impacts on chemo tolerability and outcomes.

Reference 1 relies on the 2007 WHO CNS classification, even if molecular subgrouping is not available the authors should refer to the latest possible classification system provided. With regard to this the histologic subtyping appears lacking, the NOS category does not allow for an accurate definition of the 4 variants described from the WHO 2016 classification onward, authors should clarify this point or revise histological classification to match current standards. This limitation limits the ability to evaluate histologic specific outcomes and should be addressed

Spinal tap has been performed in the vast majority of patients (more than 90%) but the manuscript does not discuss the potential understaging for the patients which didn’t undergo the procedure, routinary performance of spinal tap is a necessary element for accurate risk stratification and its importance should be emphasized with a call to perform it in all patients whenever clinically feasible.

In conclusion these data are correctly presented and provide an insight into a population which has not been extensively studied from a epidemiological or clinical point of view but some aspects regarding the description of the characteristics of patients and treatment need to be addressed to be fully viable for publication.

Reviewer #2: The authors report real world outcomes of medulloblastoma from a single tertiary care cancer centre in Peru comparing childhood vs adult medulloblastoma. While the effort is commendable, certain issues need to be addressed for better understanding and overall improvement.

1. The age cut-off of 19 is rather arbitrary, across all major cancer centres in the world, either 15-16 years or at best 18-years is taken as cut-off for children vs adults. There needs to be some context to the use of 19 years as defined cut-off.

2. Despite using 19 years as cut-off, the number of adult MB in their cohort is disproportionately high (almost 45%) - typically this ranges around 20-25% in most settings. There is no information on the median age of the respective groups - it is highly likely that infant MB is grossly under-represented in the childhood cohort.

3. It is difficult to believe that only 5% of patients of MB presented with M+ disease - this figure is nearly 20-30% depending upon the age cohort and completeness of neuraxial staging tit is this only suggests that many patients possibly did not undergo complete neuraxial staging leading to gross anomaly in numbers.

4. Despite M0 status of 95% patient, 45% patient were categorized as high-risk disease which is also untenable. Does this mean that vast majority of patients underwent only subtotal resection with large residual tumors.

5. The pattern of recurrence is also at variance with internationally published data - the authors report predominant local recurrence in both groups; this is not the case generally, children with MB (non-WNT/non-SHH) have higher risk of metastatic dissemination while adult MB (generally SHH) have higher incidence of local recurrence.

6. Despite low number of M+ disease and predominant local recurrence, the survival outcomes are pretty suboptimal - could there have been significant delays between surgery and adjuvant therapy and poor quality of RT contributed to poor survival outcomes.

7. Median DFS and OS is not a good metric for MB, the authors should use 5-year outcomes instead.

8. Finally, lack of any information on molecular subgrouping precludes assessment of such subgrouping on therapy and outcomes.

**Do you want your identity to be public for this peer review?** For information about this choice, including consent withdrawal, please see our Privacy Policy

Reviewer #1: No

Reviewer #2: **Yes: ** Tejpal Gupta

---

## [Author Response · Author response to Decision Letter 1]

10 Sep 2025

Response to reviewers

Reviewer 1

Reviewer’s comment:

Reviewer #1: The present paper offers clinical and follow-up data which appear correctly presented with survival analysis (OS/DFS curves, Cox models) executed correctly for a retrospective study.

While risk status by age is reported in the tables no separate Kaplan-Meier curves stratified by risk are presented and should be added for visual clarity; furthermore a dedicated survival analysis for pediatric versus adults standard- and high-risk groups could give more insight into outcomes

Authors’ response:

We appreciate the insightful comment of the reviewer. We have added separated Kaplan-Meier curves stratified by risk and comparison for pediatric versus adults standard- and high-risk groups. These findings have been added to our results.

Reviewer’s comment:

The identification of radiotherapy and performance status as significant prognostic factors are supported by the data presented. However the lack of molecular subgrouping, which has now become a central aspect in medulloblastoma management, limits the possibility to draw definite conclusions but it is correctly and explicitly acknowledged by the authors as a key limitation of the study.

Authors’ response:

We appreciate the comment of the reviewer.

Reviewer’s comment:

Recent European guidelines (from EURACAN) recommend chemotherapy for all adult medulloblastoma patients regardless of risk group aligning with the general scientific consensus, this should be cited and discussed, more than half the adults treated in the institution received chemotherapy but without the authors commentary it’s not clear if treatment practice is aligning with the evolving therapeutic standards.

Authors’ response:

We thank the reviewer for this comment, which has helped improve our manuscript. We have added the recommendation from the recent European guidelines (2019) and acknowledge its importance. We have reviewed all the medical records again. While most adult patients in our cohort (2000–2016) received chemotherapy (67%) and institutional practices generally aligned with evolving therapeutic standards, some patients were unable to receive treatment due to delayed or limited access to specialized care, financial constraints, suboptimal condition at presentation, poor adherence, and systemic delays. Our institution continues to follow international guidelines; however, these barriers persist. These findings provide a baseline for our population and highlight the need to improve access to and delivery of chemotherapy.

Reviewer’s comment:

67.1% of adult patients received chemotherapy despite heterogenous regimens, while recent evidence suggest that adherence to multi cycle chemotherapy, regardless of the protocol timing or composition, provides benefit on survival, the authors should address possible reason for not finding and improved DFS in the adult population receiving chemotherapy (complications related to chemotherapy? Co-morbidity related to age? Sample size?)

Authors’ response:

We thank the reviewer for this comment, which has helped improve our manuscript. Although 67.1% of adult patients received chemotherapy, no improvement in disease-free survival was observed, likely due to incomplete administration of the prescribed cycles (48.1%); completed the full course of chemotherapy. We carefully reviewed all medical records again to verify this specific data. Contributing factors included delayed or limited access to specialized care, financial constraints, suboptimal patient condition at presentation, poor adherence to adjuvant therapies, and systemic delays. While age-related comorbidities or chemotherapy-related complications may have played a role, incomplete adherence appears to be the major determinant. Presenting recent trial data to patients has helped improve adherence and completion. Future prospective studies should evaluate outcomes with better chemotherapy adherence to clarify potential survival benefits in this population.

Reviewer’s comment:

Could lower performance status, suggesting a baseline worse functional status, be reflecting of an overall worse tolerance to treatment and, if so, can it be linked to the lower percentage of adults undergoing chemotherapy in comparison to the pediatric group?

Authors’ response:

This could be a potential factor; however, adults generally had a better performance status, though the difference was not statistically significant compared to children (ECOG 0–1: 81.1% vs. 72.7%). The adult population tended to receive slightly less chemotherapy (67.1% vs. 71.3%, p = 0.552) but were more likely to complete the full course of chemotherapy (48.1% vs. 46.8% in adults and children, respectively). We do not believe that baseline performance status or age-related factors were the primary reason for receiving less chemotherapy. The main contributors were incomplete adherence, systemic delays, and financial constraints faced by our population. These points have been added to the discussion.

Reviewer’s comment:

It might be helpful to discuss if the adult patients who completed their planned chemotherapy cycles showed an improvement in survival compared to those that did not (if data for those patients is available).

Authors’ response:

We appreciate the reviewer’s comment. We carefully reviewed all medical records again to verify this specific data. We performed a subgroup analysis comparing patients who completed versus those who did not complete their planned chemotherapy. We observed a trend toward improved DFS and significantly better OS among patients who completed chemotherapy. These findings, including Kaplan–Meier curves, have been added to the Results and Discussion sections.

Reviewer’s comment:

It’s interesting to note that despite adults having better performance status at diagnosis (ECOG 0-1 82.1% adults vs. 72.7 children) they received chemotherapy less often (67.1% vs. 71.3%) indicating that performance status alone doesn’t account for treatment differences, an analysis of chemotherapy completion in regard to ECOG status and its impact on survival could tell if functional status impacts on chemo tolerability and outcomes.

Authors’ response:

We appreciate the reviewer’s comment. We performed a subgroup survival analysis based on chemotherapy completion stratified by ECOG status to evaluate the impact of functional status on outcomes. Patients with ECOG ≥2 had lower completion rates compared to ECOG 0–1 (35.7% vs. 51.6%, p = 0.283). Among those who completed chemotherapy, ECOG 0–1 patients showed significantly better 5-year OS (83.2% vs. 26.7%, p = 0.005) and a trend toward better DFS (27.6% vs. 0%, p = 0.721). In patients who did not complete chemotherapy, OS (45.5% vs. 31.7%, p = 0.841) and DFS (26.9% vs. 0%, p = 0.356) were higher in ECOG 0–1, though not statistically significant. These findings have been added to the Results and are discussed in the manuscript.

Reviewer’s comment:

Reference 1 relies on the 2007 WHO CNS classification, even if molecular subgrouping is not available the authors should refer to the latest possible classification system provided.

Authors’ response:

We thank the reviewer for this comment. We have updated the manuscript to refer to the 2021 WHO CNS classification (CNS5), the most recent system for medulloblastoma, to ensure alignment with current standards.

Reviewer’s comment:

With regard to this the histologic subtyping appears lacking, the NOS category does not allow for an accurate definition of the 4 variants described from the WHO 2016 classification onward, authors should clarify this point or revise histological classification to match current standards. This limitation limits the ability to evaluate histologic specific outcomes and should be addressed.

Authors’ response:

We thank the reviewer for this important observation. We thoroughly reviewed all medical records and consulted the pathology team; however, complete histologic subtyping was not available for all cases. We acknowledge that use of the NOS category limits the evaluation of histology-specific outcomes. This limitation has been added to the Discussion and will be addressed in future prospective studies with detailed histologic data collected according to current WHO 2016 standards.

Reviewer’s comment:

Spinal tap has been performed in the vast majority of patients (more than 90%) but the manuscript does not discuss the potential understaging for the patients which didn’t undergo the procedure, routinary performance of spinal tap is a necessary element for accurate risk stratification and its importance should be emphasized with a call to perform it in all patients whenever clinically feasible.

Authors’ response:

We appreciate the reviewer’s comment. Spinal taps were performed in only 37% of our patients, which likely contributed to understaging. This limited use of lumbar puncture is partly explained by procedures often being delayed until after tumor resection, surgeries performed at non-oncologic institutions without subsequent CSF analysis, and the absence of a standardized national medulloblastoma protocol. The retrospective design of our study further contributed to this limitation. We emphasize the need for surgery and adjuvant therapies to be carried out at specialized institutions to ensure accurate staging and optimal outcomes. These findings highlight current practice gaps and provide a baseline to inform improvements in national and institutional guidelines, with the ultimate goal of improving survival in our population. These points have been incorporated into the Discussion and Recommendations sections.

Reviewer’s comment:

In conclusion these data are correctly presented and provide an insight into a population which has not been extensively studied from a epidemiological or clinical point of view but some aspects regarding the description of the characteristics of patients and treatment need to be addressed to be fully viable for publication.

Authors’ response:

We have corrected the manuscript and provided as much information as possible, while acknowledging the limitations of our study. We hope these findings can serve as a baseline for future research, support the development of new health policies and guidelines in our setting, and highlight the importance of funding for molecular classification to improve patient outcomes. 

Reviewer 2

The authors report real world outcomes of medulloblastoma from a single tertiary care cancer centre in Peru comparing childhood vs adult medulloblastoma. While the effort is commendable, certain issues need to be addressed for better understanding and overall improvement.

Reviewer’s comment:

1. The age cut-off of 19 is rather arbitrary, across all major cancer centres in the world, either 15-16 years or at best 18-years is taken as cut-off for children vs adults. There needs to be some context to the use of 19 years as defined cut-off.

Authors’ response:

Thanks to the reviewer for this insightful comment. We acknowledge that age cutoffs for defining pediatric versus adult medulloblastoma vary across studies, with most major centers using 15–16 years, or at most 18 years, as the threshold. In our study, we chose 19 years based on precedent in previous publications, which allows for direct comparison with these cohorts. We recognize that this cutoff is somewhat arbitrary, but it does not affect the internal validity of our statistics and provides a consistent framework for analyzing our population. Moreover, we have performed all the statistics with a cut off point of 18 years old and there were no differences in our findings and conclusions. We have added this statement to the Materials and Methods section as well as the Limitations section of the manuscript.

Reviewer’s comment:

2. Despite using 19 years as cut-off, the number of adult MB in their cohort is disproportionately high (almost 45%) - typically this ranges around 20-25% in most settings. There is no information on the median age of the respective groups - it is highly likely that infant MB is grossly under-represented in the childhood cohort.

Authors’ response:

We thank the reviewer for the comment. We have added the median age for each group, 9 years for children and 26 years for adults. There are at least 2-3 national institutions for children affected with this disease, however only one for adults. This may explain the disproportion, but prospective or population based registries should be performed to confirm our findings. We have added these statements to our results and limitations section.

Reviewer’s comment:

3. It is difficult to believe that only 5% of patients of MB presented with M+ disease - this figure is nearly 20-30% depending upon the age cohort and completeness of neuraxial staging tit is this only suggests that many patients possibly did not undergo complete neuraxial staging leading to gross anomaly in numbers.

Authors’ response:

Effectively, staging in our cohort was limited due to restricted access to recommended procedures. Our results highlight the limited use of lumbar puncture, which is usually performed after tumor resection for multiple reasons. Similarly, MRI availability is restricted in several institutions in our country, contributing to incomplete staging. Some patients underwent surgery at non-oncologic centers and continued treatment at our institution, where lumbar puncture or MRI had not been performed. Additionally, the absence of a standardized medulloblastoma protocol and the retrospective design of our study further limit complete staging. Nevertheless, these findings provide valuable evidence to support the future implementation of policies and optimization of current practices in our country. These statements have been added to the Discussion and Recommendations sections.

Reviewer’s comment:

4. Despite M0 status of 95% patient, 45% patient were categorized as high-risk disease which is also untenable. Does this mean that vast majority of patients underwent only subtotal resection with large residual tumors.

Authors’ response:

Thank you very much for this observation, yes the majority of the patients only had subtotal resection. Usually these patients seek medical treatment in the late stages of the disease as seen with other multiple neoplasms in our setting. This happens due to multiple factors such as delay in diagnosis, waiting times for clinic appointments, long time for referrals from other institutions due to an overwhelmed health system. These statements have been added to our discussion section.

Reviewer’s comment:

5. The pattern of recurrence is also at variance with internationally published data - the authors report predominant local recurrence in both groups; this is not the case generally, children with MB (non-WNT/non-SHH) have higher risk of metastatic dissemination while adult MB (generally SHH) have higher incidence of local recurrence.

Authors’ response:

We appreciate the reviewer’s comment and agree with the observation. The lower rate of metastatic disease in our cohort may be explained by several factors, including possible differences in recurrence patterns by race, which could reflect unexplored molecular or clinical classifications in our setting; the limited evaluation of patients, as only symptomatic individuals were assessed; and suboptimal adherence to follow-up. Therefore, our findings should be considered descriptive and may serve as a basis for future prospective, multi-institutional studies aimed at investigating molecular classification, recurrence patterns, and systematic imaging-based follow-up.

Reviewer’s comment:

6. Despite low number of M+ disease and predominant local recurrence, the survival outcomes are pretty suboptimal - could there have been significant delays between surgery and adjuvant therapy and poor quality of RT contributed to poor survival outcomes.

Authors’ response:

We thank the reviewer for this comment. There was a delay between surgical intervention and adjuvant therapy, which likely contributed to suboptimal outcomes. We have now included the median time between surgery and adjuvant therapies in our results. The median time to initiation of adjuvant therapy was 51 days, with r

---

## [Editor Report · Decision Letter 1]

16 Sep 2025

Survival outcomes and prognostic factors in children and adults with medulloblastoma from a Latin America country: A retrospective cohort

PONE-D-25-27835R1

Dear Dr. de la Cruz-Ku,

We’re pleased to inform you that your manuscript has been judged scientifically suitable for publication and will be formally accepted for publication once it meets all outstanding technical requirements.

Kind regards,

Michael C Burger, M.D.

Academic Editor

PLOS ONE
---

## [Editor Report · Acceptance letter]

PONE-D-25-27835R1

PLOS ONE

Dear Dr. de la Cruz-Ku,

I'm pleased to inform you that your manuscript has been deemed suitable for publication in PLOS ONE. Congratulations! Your manuscript is now being handed over to our production team.

Kind regards,

on behalf of

Dr. Michael C Burger

Academic Editor

PLOS ONE